# Evaluation of Hepcidin Level in COVID-19 Patients Admitted to the Intensive Care Unit

**DOI:** 10.3390/diagnostics12112665

**Published:** 2022-11-02

**Authors:** Marco Ciotti, Marzia Nuccetelli, Massimo Pieri, Carlo Maria Petrangeli, Alfredo Giovannelli, Terenzio Cosio, Luigi Rosa, Piera Valenti, Francesca Leonardis, Jacopo Maria Legramante, Sergio Bernardini, Elena Campione, Marilena Minieri

**Affiliations:** 1Virology Unit, Polyclinic Tor Vergata, Viale Oxford 81, 00133 Rome, Italy; 2Department of Laboratory Medicine, Polyclinic Tor Vergata, Viale Oxford 81, 00133 Rome, Italy; 3Department of Experimental Medicine, University of Tor Vergata, 00133 Rome, Italy; 4Intensive Care Unit, Polyclinic Tor Vergata, Viale Oxford 81, 00133 Rome, Italy; 5Dermatology Unit, Department of Systems Medicine, Polyclinic Tor Vergata, 00133 Rome, Italy; 6Department of Public Health and Infectious Diseases, University of Rome “La Sapienza”, 00185 Rome, Italy; 7Emergency Medicine, Polyclinic Tor Vergata, Viale Oxford 81, 00133 Rome, Italy; 8Emerging Technologies Division (ETD) of the International Federation Clinical Chemistry and Laboratory Medicine (IFCC), 20159 Milan, Italy

**Keywords:** COVID-19, hepcidin, ICU

## Abstract

Coronavirus disease 2019 (COVID-19) presents a clinical spectrum that ranges from a mild condition to critical illness. Patients with critical illness present respiratory failure, septic shock and/or multi-organ failure induced by the so called “cytokine storm”. Inflammatory cytokines affect iron metabolism, mainly inducing the synthesis of hepcidin, a hormone peptide not routinely measured. High levels of hepcidin have been associated with the severity of COVID-19. The aim of this study was to analyze, retrospectively, the levels of hepcidin in a group of COVID-19 patients admitted to the intensive care unit (ICU) of the Policlinico Tor Vergata of Rome, Italy. Thirty-eight patients from November 2020 to May 2021 were enrolled in the study. Based on the clinical outcome, the patients were assigned to two groups: survivors and non-survivors. Moreover, a series of routine laboratory parameters were monitored during the stay of the patients in the ICU and their levels correlated to the outcome. Statistical differences in the level of hepcidin, D-dimer, IL-6, LDH, NLR, neutrophils level, CRP, TNF-α and transferrin were observed between the groups. In particular, hepcidin values showed significantly different median concentrations (88 ng/mL vs. 146 ng/mL) between survivors and non-survivors. In addition, ROC curves analysis revealed sensitivity and specificity values of 74% and 76%, respectively, at a cut-off of 127 (ng/mL), indicating hepcidin as a good biomarker in predicting the severity and mortality of COVID-19 in ICU patients.

## 1. Introduction

Coronavirus disease 2019 (COVID-19), caused by the recently uncovered human coronavirus SARS-CoV-2 [1], presents a clinical picture that ranges from a mild condition to critical illness [2]. Patients with critical illness present respiratory failure, septic shock and/or multi-organ failure induced by the so defined “cytokine storm” [3]. This exaggerated inflammatory response is characterized by increased level of several cytokines including IL-2, IL-6, IL-7 and interferon-c inducible protein-10, macrophage inflammatory protein 1-α, tumor necrosis factor-α (TNF-α), monocyte chemoattractant protein 1 and granulocyte colony stimulating factor [4]. Inflammatory cytokines affect iron metabolism mainly inducing the synthesis of hepcidin [5], a 25-amino acid peptide synthesized primarily in liver cells, which is the principal regulator of iron absorption and its distribution to tissues. Serum ferritin is also an important marker of iron status and a measure of inflammation. High levels of ferritin are associated with high mortality in COVID-19 patients and are reflected in disordered iron metabolism [6].

Hepcidin modulates cellular iron export to plasma and extracellular fluid through ferroportin, which acts as hepcidin receptor and cellular iron exporter in vertebrates. The levels of hepcidin are homeostatically regulated by iron and erythropoietic activity. An excess of iron stimulates the production of hepcidin that in turn inhibits the absorption of iron from the intestine. On the contrary, hepcidin is suppressed in iron deficiency, allowing dietary iron absorption and replenishment of iron storage [7,8]. In inflammation and infection, an increase in the hepcidin level can be observed, likely linked to the host defense mechanism to subtract iron from the infecting microorganisms [9]. Hepcidin sequesters iron mainly through macrophages [10]. This phenomenon is well known in bacterial infections, while its role in viral infections is unclear. Iron metabolism has implications on the function of the immune system. Lymphocytes are primed by contact with the antigen presenting cells, and they need iron to mount a robust and effective cellular and humoral response.

Despite the fundamental role of hepcidin in the regulation of iron metabolism, hepcidin is not routinely measured in the clinic laboratory. Preliminary studies have shown that high levels of hepcidin in COVID-19 patients are associated with the severity of the disease [5,11]. Similarly, low levels of serum iron have been correlated with severity and mortality of COVID-19 [12] and severe hypoxemia in intensive care unit (ICU) patients [13].

Given this background, the study aimed at retrospectively analyzing the levels of hepcidin in a group of COVID-19 patients admitted to the ICU of the Policlinico Tor Vergata of Rome, Italy. The levels of hepcidin were correlated to the outcome of the disease.

## 2. Materials and Methods

### 2.1. Study Group

Thirty-eight patients admitted to the ICU from November 2020 to May 2021 because of severe pneumonia caused by SARS-CoV-2 were enrolled in the study. Based on the clinical outcome, the patients were assigned to two groups: survivors and non-survivors. The median age in the survivors’ group was 65.5 years old, while in the non-survivors was 73.5 years old. A total of 18 patients belonged to the survivors’ group (13 males; 5 females) and 20 to the non-survivors’ group (10 males; 10 females). A series of laboratory parameters were monitored during the stay of the patients in the ICU and their levels correlated with the clinical outcome. The samples were collected during the first week of admission in the ICU.

The study was approved by the local Ethical Committee of the Azienda Ospedaliera Policlinico Tor Vergata, under the protocol number 42.20/2020.

### 2.2. Measurement of Hepcidin Level

Hepcidin level was measured using the “Intrinsic Hepcidin IDx™ ELISA kit” (Intrinsic Lifesciences, San Diego, CA, USA), which is a competitive immunoenzymatic assay based on a monoclonal antibody (mAb) with high affinity to the N-terminus of hepcidin-25. This antibody also binds low abundance, N-terminus isomers of hepcidin-25, but with lower affinities. The assay was used for the quantitative measurement of hepcidin in human serum. Hepcidin-25 in the test specimen competes with a biologically active biotinylated human hepcidin-25 tracer for the binding to high affinity anti-hepcidin-25 N-terminal specific mAbs coated on the plate. Briefly, standards (ranging from 0 to 1000 ng/mL), controls and samples were incubated with biotinylated hepcidin-25 tracer for 60 min. After washing, a conjugate solution labeled with horseradish peroxidase (HRP) was added to each well and incubated for 30 min. Then, the HRP substrate (tetramethylbenzidine; TMB) was dispensed and the reaction stopped after 15 min using an acidic solution. The absorbance was read on a spectrophotometer at 450 nm (DAS srl, Rome, Italy). Optical densities are inversely proportional to hepcidin-25 concentration in the samples. This test is CE approved.

### 2.3. Measurement of Other Laboratory Parameters

Hematological and biochemical parameters were measured on blood, serum and plasma samples collected at the ICU. White blood cells (WBCs), eosinophils, lymphocytes, neutrophil count, hemoglobin and platelets were measured by using an automated hematological analyzer (Dasit Sysmex, Milan, Italy). The neutrophil-to-lymphocyte ratio (NLR) and Lymphocyte-Neutrophil ratio (LNR) was also determined. Serum levels of high-sensitivity C-reactive protein (hsCRP; reference range 0–5 mg/L) were measured by using an immunoturbidimetric method (Abbott Diagnostics, Milan, Italy). Serum levels of Tumour Necrosis Factor alpha (TNF-α) (reference range: 0–12.4 pg/mL) were measured using an enzyme-linked immunosorbent assay (ELISA) technique (DRG, International Instruments GmbH, Marburg, Germany). Serum levels of Interleukin-6 (IL-6) (reference range: 0–50 pg/mL) were measured using the chemiluminescence method (IMMULITE 2000 instrument, Siemens, Milan, Italy). Serum levels of ferritin (reference range: 21.81–274.66 ng/mL) were measured using the chemiluminescence method; lactate dehydrogenase (LDH) (reference range: 125.00–220.00 U/L), serum iron (reference range: 50–175 µg/dL) and transferrin (reference range: 174–364 mg/dL) by a colorimetric method (Alinity Instrument, Abbott, Milan, Italy). Plasma fibrinogen concentrations (reference range: 200–400 mg/dL) and plasma D-dimer levels (reference range: 0–500 ng/mL) were measured by ACL-TOP 500 instrumentation (Werfen, Milan, Italy).

### 2.4. Statistical Analysis

The statistical analysis was performed using both the histogram and the Kolmogorov–Smirnov test of normality (*p* value < 0.05) to verify the normal distribution of data. In the case of more than two variables, analysis of variance with Bonferroni post-hoc test was performed, or the *t*-test in the case of two variables.

To assess differences between groups, the Kruskal–Wallis test (variables with more than two categories) and Mann–Whitney U test (variables with two categories), were used. A *p*-value < 0.05 was considered statistically significant.

The sensitivity and specificity of the biochemical parameters was evaluated by ROC curve analysis performed in survivors vs. non-survivors COVID-19 patients.

All statistical analyses were performed using MedCalc Version 18.2.18 (MedCalc Software Ltd., Ostend, Belgium).

## 3. Results

The study included 18 patients in the survivors’ group (13 males and 5 females) and 20 patients in the non-survivors’ group (10 males and 10 females). They were homogeneous regarding age; median age 65.5 years (I.R. 58–74) in the survivors’ group, and 73.5 years (I.R. 66.5–79.5) in the non-survivors’ group (*p* > 0.05). Several clinical biochemical tests were evaluated and the results are shown in Table 1.

Statistically significant differences in the level of hepcidin, D-dimer, IL-6, LDH, neutrophils count, NLR, CRP, TNF-α and transferrin (Mann–Whitney U test; *p* < 0.05) were observed between the survivors and non-survivors’ groups (Table 1 and Figure 1); no statistically significant differences were observed between the two groups of patients for AST/GOT, leukocyte count, eosinophils count, ferritin, fibrinogen, hemoglobin, lymphocytes count, lipase, platelets and sideremia, Table 1. In particular, hepcidin values show significantly different median concentrations (88 ng/mL vs. 146 ng/mL) between the survivors and non-survivors’ group, respectively (*p* < 0.005). In Figure 1, the median values with the 25th and 75th percentile and the individual data obtained from the patients are reported.

Finally, ROC curve analyses were performed on the laboratory parameters with significant differences. Data obtained are reported in Table 2. LDH and IL-6 showed the best sensitivity (95%) and specificity (100%) values, respectively. Results for all other routine parameters are in line with the current literature. Hepcidin showed good sensitivity and specificity values of 74% and 76%, respectively, at a cut-off of 127 (ng/mL) with an AUC of 0.813 (Figure 2). Thus, a value of hepcidin lower than 127 (ng/mL) at hospital admission is suggestive of a favorable prognosis, also used alongside other biomarkers.

## 4. Discussion

Since its first appearance at the end of 2019, COVID-19 spread rapidly throughout the world, causing millions of deaths [14]. Although most of the patients present mild symptoms and have a good prognosis, some patients develop a more severe clinical condition including severe pneumonia, acute respiratory distress or multiorgan failure with death [15]. In these critically ill patients, a massive inflammatory response with the release of numerous cytokines, known as a “cytokine storm”, can be observed [16]. This inflammatory response alters a series of hematological and biochemical parameters.

Recent studies investigated the potential role of hepcidin (an important regulator of iron metabolism) as predictor of disease severity and mortality in COVID-19 patients [5,11]. Indeed, elevated serum levels of hepcidin were found in patients with severe disease or unfavorable outcome.

In this regard, in this study, we evaluated the routine hematological and biochemical profile as well as the hepcidin level, in a group of COVID-19 patients with severe pneumonia admitted to the ICU

Our analyses showed that there are statistically significant differences in the level of some parameters between survivors and non-survivors. In particular, higher levels of D-dimer, IL-6, LDH, NLR, CRP TNF-α and transferrin were observed in the non-survivors compared to the survivors.

Elevated levels of D-dimer (a biomarker used to detect thrombosis) are linked to poor prognosis and need for ICU care in COVID-19 patients [17,18]. A recent meta-analysis [19] reported that patients who died had a higher level of D-dimer compared to those who survived, leading to conclusions in agreement with our results.

LDH is a recognized biomarker of COVID-19 severity and mortality [20]. This observation was confirmed by a recent meta-analysis that also denoted how LDH is much higher in ICU vs. non-ICU patients and in non-survival patients vs. survival patients. Thus, it can be used as a predictor of survival [21]. The importance of the prognostic value of LDH level in COVID-19 patients has been reconfirmed by a large retrospective study published at the beginning of 2022 [22]. Univariate regression analysis showed that patients with elevated LDH level had a higher risk of in-hospital death than those with normal or decreased level of LDH. In addition, the Kaplan–Meier curves showed that elevated LDH was associated with in-hospital death [22]. In our study, the higher level of LDH in the non-survivors’ group vs. survivors’ group supports this last consideration.

Numerous scientific articles have reported on the association of IL-6 and TNF-α with the severity and mortality of COVID-19 [4,23,24,25,26]. Elevated levels of these two biomarkers significantly increase the risk of disease severity and mortality as it has been recently reviewed in a study which included a large sample size. The non-survivors patients presented a marked difference in the level of IL-6 compared to the survivors [27]. A similar trend was observed for the TNF-α level [28]. Our data confirm that elevated levels of IL-6 and TNF-α are associated with an unfavorable prognosis.

Moreover, increased NLR and neutrophils count are predictors of poor prognosis and severity of disease [29,30] as well as elevated levels of CRP [31,32]. Our results confirm these observations as shown by the higher levels of these laboratory indexes in the non-survivors’ group.

Lower levels of transferrin have been associated to a heightened inflammatory response and disease progression [33]. In our study, group non-survivors presented median lower transferrin levels compared to the survivors suggesting that low level transferrin could be a predictor of increased disease severity [33].

Recent studies have investigated the role of hepcidin as predictor of disease severity and mortality in COVID-19 patients [5,11,34]. In our work, all patients admitted to the ICU because of their severe pneumonia showed elevated levels of hepcidin and these levels were even higher in those patients who did not survive and required an invasive mechanical ventilation.

Finally, ROC curve analyses showed that LDH and IL-6 have the best sensitivity and specificity, respectively. Nonetheless, hepcidin showed a sensitivity of 74% and a specificity of 76%, at a cut-off value of >127 ng/mL, suggesting that hepcidin measurement is a useful biomarker for predicting the severity and the outcome of COVID-19 in ICU patients. Furthermore, hepcidin could be used to define an algorithm along with other recent novel inflammatory biomarkers such as serum amyloid A protein (SAA) and mid-regional proadrenomedullin (MR-proADM) that have been recently evaluated for predicting mortality in COVID-19 patients [35,36]. Higher levels of SAA were found in patients who died compared to the survivors [35]. Similarly, patients with a level of MR-proADM higher than 1105 nmol/L had a three-fold higher risk of mortality [36].

Lastly, although, they remain important indicators of disease severity [33,37], our analysis did not reveal statistically significant differences in the levels of fibrinogen, AST, ferritin, leucocytes, lymphocytes and sideremia between the survivors and non-survivors groups.

A limitation of the study is the small sample size that might reduce the power of the results. However, this investigation confirms the usefulness of some biomarkers in predicting the severity and mortality of COVID-19 and brings important information about the utility of hepcidin measurement in COVID-19 patients admitted to the ICU. All patients with an elevated hepcidin level had an unfavorable outcome.

## Figures and Tables

**Figure 1 diagnostics-12-02665-f001:**
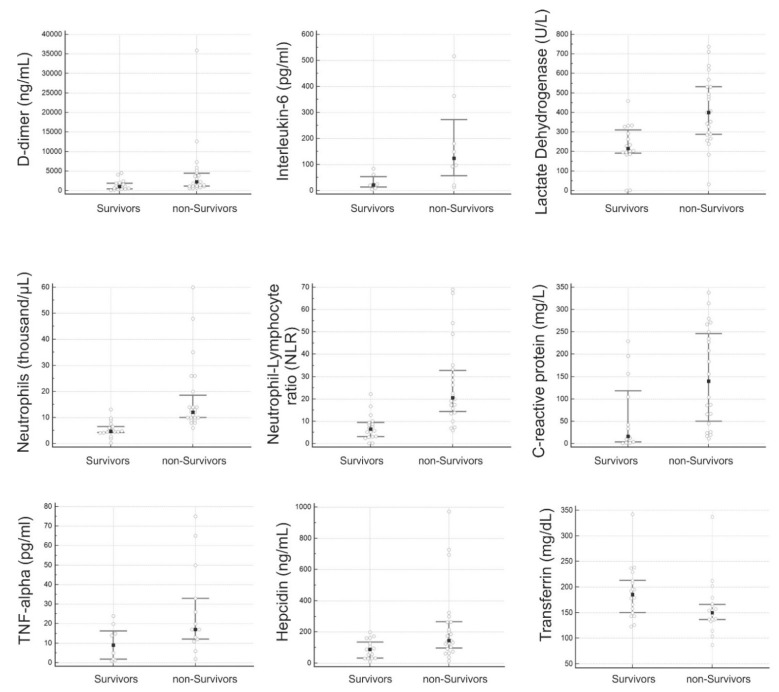
Serum levels of laboratory parameters significantly different in non-survivors’ and survivors’ groups (Mann–Whitney U test).

**Figure 2 diagnostics-12-02665-f002:**
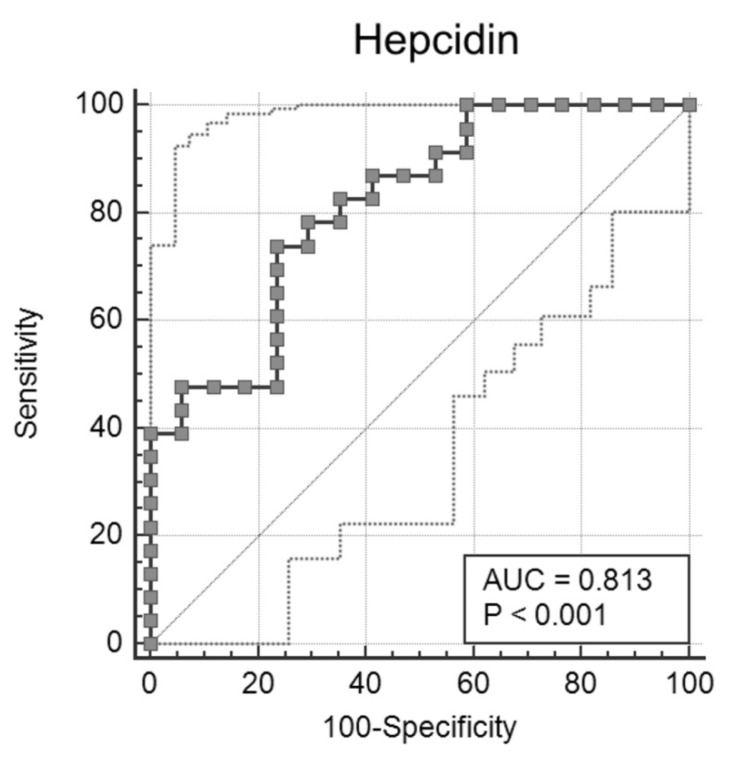
ROC curve analysis of hepcidin in non-survivors’ and survivors’ groups.

**Table 1 diagnostics-12-02665-t001:** Statistical analysis of laboratory parameters. For each study group are indicated: minimum, maximum, median and 95% confidence interval. A *p*-value higher than 0.05 (>0.05) is not statistically significant.

	Survivors				Non-Survivors				
	Min	Max	Median	95% CI	Min	Max	Median	95% CI	*p*-Value
Aspartate Aminotransferase (U/L)	15	59	29	21.06 to 40.08	12	1876	31	23.34 to 53.65	0.455
D-dimer (ng/mL)	80	4507	1062.5	485.9 to 1770.9	581	35,873	2223	1156.41 to 3802.27	0.014
Eosinophils (thousand/μL)	0	1.15	0.0450	0.020 to 0.10	0	0.81	0.01	0.0 to 0.11	0.86
Hepcidin (ng/mL)	10.2	198.1	88	33.03 to 126.69	13.08	971.44	144.96	111.67 to 245.59	0.005
Ferritin (ng/mL)	198	1315	596.5	332.27 to 763.80	434.01	11,853	1335	487.57 to 4555.79	0.17
Fibrinogen (mg/dL)	95	919	450	367.38 to 556.62	36	1144	508	392.0 to 611.55	0.438
Hemoglobin (ng/dL)	8	13.9	10.95	9.61 to 12.02	7.6	15.3	9.7	8.7 to 11.03	0.52
Interleukin-6 (pg/mL)	5	84.3	22	7.48 to 73.10	13	517	124	19.48 to 392.95	0.032
Lactate Dehydrogenase (U/L)	184	974	231	200.47 to 329.44	185	737	400	292.48 to 518.44	0.003
Leucocyte (thousand/μL)	0.29	13	3.21	0.812 to 6.603	0.26	49	1.01	0.547 to 9.651	0.9895
Lymphocytes (thousand/μL)	0.27	21.7	0.915	0.60 to 1.39	0.26	2.59	0.58	0.56 to 1.047	0.168
Lipase (U/L)	9	137	30	24.26 to 44.42	7	328	28	23.0 to 59.49	0.849
Neutrophils (thousand/μL)	2.83	21.69	5.045	4.52 to 7.19	0.41	47.88	12.12	9.86 to 13.80	0.0001
Neutrophil-Lymphocyte ratio (NLR)	0.204	36.15	7.311	4.01 to 9.87	6.198	68.345	20.5	16.56 to 28.13	0.0001
C-reactive protein (mg/L)	2.3	228.8	33.9	7.34 to 157.62	11.2	337.6	148.35	66.93 to 236.06	0.015
Platelets (thousand/μL)	59	485	225.5	172.31 to 304.24	55	439	178	144.48 to 242.20	0.27
Tumor Necrosis Factor-α (pg/mL)	0.97	24	9	1.448 to 19.31	2	75	17	11.89 to 22.02	0.0438
Transferrin (mg/dL)	123	342	185.5	153.17 to 211.01	87	337	150	136.78 to 163.87	0.0245
Sideremia (µg/dL)	28	137	55.5	43.38 to 69.20	21	165	72	40.17 to 108.66	0.38

**Table 2 diagnostics-12-02665-t002:** ROC curve analysis of laboratory parameters in non-survivors and survivors’ groups. Sensitivity, specificity, cut-off, area under curve (AUC) and 95% confidence interval are reported.

n = 38; 18 Survived, 20 Died	D-Dimer (ng/mL)	Hepcidin (ng/mL)	IL-6 (pg/mL)	Lactate Dehydrogenase (U/L)	Neutrophils (Thousand/μL)	Neutrophil Lymphocyte Ratio (NLR)	C-Reactive Protein (mg/L)	Tumor Necrosis Factor-α (pg/mL)	Transferrin (mg/dL)	Sideremia (µg/dL)
Sensitivity (%)	52	74	75	95	91	82	82	80	68	50
Specificity (%)	83	76	100	56	78	83	61	43	72	78
Cut-off	>1981	>127	>84	>234	>7.62	>12.75	>41.5	>17.35	<157	>70
Area under the ROC curve (AUC); 95% Confidence interval	0.726; 0.564 to 0.853	0.813; 0.659 to 0.919	0.830; 0.553 to 0.970	0.779; 0.617 to 0.895	0.853; 0.707 to 0.944	0.850; 0.704 to 0.942	0.748; 0.573 to 0.879	0.557; 0.302 to 0.792	0.716; 0.545 to 0.852	0.583; 0.412 to 0.741

## Data Availability

Data are reported within the article.

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
