# Peer review of "Evaluation of Hepcidin Level in COVID-19 Patients Admitted to the Intensive Care Unit"

_diagnostics, 2022, doi:10.3390/diagnostics12112665_

Round 1

Reviewer 1 Report

The authors show convincing evidence that, although not superior to LDH and IL-6, Hepcidin has a high sensitivity (74%) and a specificity (76%) at a cut-off value of >127 ng/ml for predicting the severity and the outcome of COVID-19 in ICU patients. As a modulator of cellular iron export to plasma, hepcidin is indirectly reflected in serum iron and transferrin saturation levels, which have been consistently demonstrated as good predictors of the severity and mortality of COVID-19 and severe hypoxemia in intensive care unit (ICU) patients. Therefore, these parameters should also be included in the Statistical analysis of laboratory parameters. From a clinical practical point of view, and given the simplicity and low cost of serum iron parameters relative to hepcidin, a comparison of the ROC curves should be provided, and the eventual superiority of hepcidin as a marker in predicting the severity and mortality of COVID-19 in ICU patients should be demonstrated in this cohort of patients.

 Another controversial aspect in the clinical management of COVID-19 patients is the value of serum ferritin, which is extremely variable in the course of the disease, not all patients showing their peak at the same time point. Therefore, any aggregated analysis of different patients may be confounded by the time of collection, and this may be one reason why many studies are contradictory in this parameter. It is not clear what measure of hepcidin was taken in each case (Diagnosis? First week? Follow-up? Peak value?). For the purpose of uniformization authors should use some kind of stratified values. I suggest to use the peak value in the first week of hospitalization.

Author Response

The authors show convincing evidence that, although not superior to LDH and IL-6, Hepcidin has a high sensitivity (74%) and a specificity (76%) at a cut-off value of >127 ng/ml for predicting the severity and the outcome of COVID-19 in ICU patients. As a modulator of cellular iron export to plasma, hepcidin is indirectly reflected in serum iron and transferrin saturation levels, which have been consistently demonstrated as good predictors of the severity and mortality of COVID-19 and severe hypoxemia in intensive care unit (ICU) patients. Therefore, these parameters should also be included in the Statistical analysis of laboratory parameters. From a clinical practical point of view, and given the simplicity and low cost of serum iron parameters relative to hepcidin, a comparison of the ROC curves should be provided, and the eventual superiority of hepcidin as a marker in predicting the severity and mortality of COVID-19 in ICU patients should be demonstrated in this cohort of patients.

Serum iron and transferrin have been included in the statistical analysis and ROC curves performed as suggested. We found a statistically significant difference in the level of ferritin between survivors and non-survivors. Non-survivors show lower median levels of ferritin compared to the survivors. Low levels ferritin have been associated to disease progression and disease severity (Int J Infect Dis 2022; 116, 74-79).

ROC curves analysis confirms the superiority of hepcidin as predictor of severity and mortality of COVID-19 in ICU patients.

Based on the new analyses, table 1, table 2 and figure 1 have been modified accordingly.

 Another controversial aspect in the clinical management of COVID-19 patients is the value of serum ferritin, which is extremely variable in the course of the disease, not all patients showing their peak at the same time point. Therefore, any aggregated analysis of different patients may be confounded by the time of collection, and this may be one reason why many studies are contradictory in this parameter. It is not clear what measure of hepcidin was taken in each case (Diagnosis? First week? Follow-up? Peak value?). For the purpose of uniformization authors should use some kind of stratified values. I suggest to use the peak value in the first week of hospitalization.

We actually used the peak value measured in the first week of hospitalization. We clarify this point in the paragraph 2.1, Study group.

Reviewer 2 Report

Abstract 37 Delete "good" sensitivity

Intro 50_53 Consider that iron itself drives inflammation. Ferrinin drives MF cytokine elaboration.

Methods 143 specify when samples were gotten

Consider adding this reference:

Author Response

Abstract 37 Delete "good" sensitivity

Word deleted as suggested.

Intro 50_53 Consider that iron itself drives inflammation. Ferrinin drives MF cytokine elaboration.

As suggested by the reviewer, we added a sentence on ferritin as a dual marker of iron status and inflammation.

Methods 143 specify when samples were gotten

The samples were collected during the first week of admission in ICU. This sentence has been added in the paragraph 2.1 Study group, line 87-88.

Consider adding this reference:

The reference “Metallomics, 13, 2021, mfab030. Optimal serum ferritin level range: iron status measure and inflammatory biomarker” has been added as suggested and cited in the sentence about ferritin.